# Quantifying the association between ethnicity and COVID-19 mortality: a national cohort study protocol

Hajira Dambha-Miller ⬛,[1,2] Pui San Tan,[3] Defne Saatci,[3] Ashley Kieran Clift ⬛,[3] Francesco Zaccardi,[4] Carol Coupland,[5] Patrick Locufier,[6] Firoza Davies,[6] Kamlesh Khunti ⬛,[7] Simon J Griffin,[8] Julia Hippisley-Cox ⬛ [9]

**Correspondence to**
Dr Hajira Dambha-Miller; hajiradambha@doctors.org.uk

## ABSTRACT

**Introduction** Recent evidence suggests that ethnic minority groups are disproportionately at increased risk of hospitalisation and death from SARS-CoV-2 infection. Population-based evidence on potential explanatory factors across minority groups and within subgroups is lacking. This study aims to quantify the association between ethnicity and the risk of hospitalisation and mortality due to COVID-19.

**Methods and analysis** This is a retrospective cohort study of adults registered across a representative and anonymised national primary care database (QResearch) that includes data on 10 million people in England. Sociodemographic, deprivation, clinical and domicile characteristics will be summarised and compared across ethnic subgroups (categorised as per 2011 census). Cox models will be used to calculate HR for hospitalisation and COVID-19 mortality associated with ethnic group. Potential confounding and explanatory factors (such as demographic, socioeconomic and clinical) will be adjusted for within regression models. The percentage contribution of distinct risk factor classes to the excess risks seen in ethnic groups/subgroups will be calculated.

**Ethics and dissemination** The study has undergone ethics review in accordance with the QResearch agreement (reference OX102). Findings will be disseminated through peer-reviewed manuscripts, presentations at scientific meetings and conferences with national and international stakeholders.

## Strengths and limitations of this study

► The QResearch database includes a large sample that represents 18% of the English population.
► As data are based on primary care clinical entries, recording of exposures and outcomes may vary in completeness.
► Total COVID-19 cases in the cohort will be underestimated as national systematic testing is still being established in the UK.

## INTRODUCTION

Recent reports and early observational data suggest that the prevalence of COVID-19, hospitalisation and deaths are disproportionately higher among ethnic minority populations.[1–3] Potential speculative mechanisms include genetic variations, comorbidities, cultural or behavioural patterns, occupational factors, social constructs and inequalities. In the UK, people with ethnic minority backgrounds constitute 16% of the population, but the Intensive Care National Audit and Research Centre reports that over 30% of people admitted to intensive care with COVID-19 were from these backgrounds.[4 5] Furthermore, 64% of 119 National Health Service (NHS) staff who died from COVID-19 as of 30 April 2020 were from ethnic minority backgrounds.[6] A rapid review on the subject led by Khunti *et al* suggests that there are also variations in mortality within broad ethnic minority groups.[3] Most national and international data on COVID-19 are aggregated and have not reported outcome by ethnic group sufficiently to permit examination of within-group differences.

At the time of submitting this manuscript, a limited number of population-based cohort studies regarding COVID-19 have been undertaken worldwide. In the USA, a study within the largest healthcare system, the Veterans Health Administration, reports excess mortality among ethnic minorities.[7] In the UK, the availability of high-quality, large (multimillion) databases that have linkage across varying information sources are starting to examine this association further. Large primary care databases have been used, for example, to develop risk prediction models for adverse COVID-19 outcomes in the general population and to undertake large-scale epidemiological evaluations of COVID-19 risk factors and drug safety.[8] In the UK, two large cohorts have reported to date on the association between ethnicity and COVID-19-related mortality outcomes: the OpenSAFELY Collaborative of

BMJ

23 million UK records and the NHS England cohort study of over 61 million people. The OpenSAFELY consortium reported that people with black or Asian backgrounds have a higher risk of death from COVID-19 compared with people who are white.[9] Data from NHS England also showed a greater mortality risk for Asian (HR 1.3) and black (HR 1.7) backgrounds. Further analysis of these data, carried out by the Office for National Statistics (ONS), showed that the mortality risk in black men and women was 1.9 times higher than in white individuals. Contradictory to a previous study in a smaller, single-state US health system database, both studies identify a persisting excess risk of COVID-19-specific mortality after accounting for known risk factors, including age, sex, deprivation and comorbidities such as obesity, diabetes and cardiovascular disease. The causes of these excess risks are poorly understood, incompletely quantified and may vary within and between ethnic minority groups.

Previous studies have been limited by the lack of consideration of additional potential explanatory variables, such as specific medications, rarer comorbidities,[10] social differences such as household structure, or vaccination status.[9] A recent population-based study, for example, identified a significant interaction between ethnicity and ACE inhibitors/angiotensin receptor blockers for COVID-19.[11] Several smaller studies have also suggested a role of glucose-lowering medications in the ethnicity-dependent variations observed in COVID-19 outcomes.[12 13] Single-centre studies and case series are providing additional insights into the potential interaction between ethnicity and haematological disorders, such as sickle cell disease,[10] in relation to COVID-19. Population-based studies with adequate number of people from different ethnic groups that consider these wider variables are necessary to evaluate the explanatory factors surrounding this risk. Moreover, earlier studies have been limited, and larger cohorts that extend to the wider English population are needed for validation and to allow generalisability of findings. Accordingly, in this study we aim to quantify ethnic group-specific risk of SARS-CoV-2 infection, COVID-19 mortality and COVID-19 hospital admission, and examine the effect of non-modifiable (eg, age and sex) and explanatory factors on the associations between ethnicity and mortality/hospitalisation risk.

## METHODS
### Study design
This is a retrospective cohort study from electronic general practice (GP) records linked at the individual level to hospital admissions, SARS-CoV-2 testing and death registry databases.

### Data sources
We will use the QResearch national primary care database, which includes 1205 GPs with 10 million records. The sample represents 18% of the English population. QResearch holds data on a heterogeneous and representative ethnic minority population, with comprehensive information on a range of clinical and socioeconomic factors. Further details on the population included within the QResearch database and representative are updated regularly on the database website.[14] Additional databases may be included as these become available.

### Study period
We will include the period from the date of the first case of COVID-19 in the UK (24 January 2020) until death, deregistration or 30 October 2020.

### Inclusion criteria
We will include all adults aged over 18 years registered with a GP practice during the study period and for at least 12 months prior to inclusion. The GP practice registration needs to be linked to a relevant clinical system (ie, EMIS).

### Primary outcome
The primary outcome will be COVID-19-related mortality as recorded in the primary care records and linked to ONS data. Details on the extent of linkage available within the database have been reported previously and can be found in full on the QResearch website.[15] COVID-19-related mortality will be defined as the presence of U07.1 or U07.2 on death certificates or death from any cause in a patient with a confirmed positive SARS-CoV-2 test in the immediately preceding 28 days.

### Secondary outcomes
COVID-19 hospitalisation was defined as hospitalisation with confirmed or suspected COVID-19 (as per International Classification of Diseases-10 codes), or new hospitalisation in a patient with a positive SARS-CoV-2 test in the immediately preceding 14 days.

### Ethnicity definition
A self-reported ethnic group is recorded in medical records. QResearch has ethnicity records in 80% of all patients. Ethnicity will be considered in two ways: first in 9 high-level categories: white, Indian, Bangladeshi, Pakistani, other Asian, black African, black Caribbean, Chinese and 'Other'; and then, if numbers allow, further subdivided into the standard 17 categories used by ONS and include the following: English/Welsh/Scottish/Northern Irish/British, Irish, Gypsy or Irish Traveller, any other white background, white and black Caribbean, white and black African, white and Asian, any other mixed/multiple ethnic backgrounds, Indian, Pakistani, Bangladeshi, Chinese, any other Asian background, African, Caribbean, any other black/African/Caribbean background, Arab or any other ethnic group.

### Explanatory variables
We will extract data (where available) corresponding to the following potential exposure and confounding variables. The following list is indicative and correct as of 29

August 2020, but will be kept under review as new information arises in the literature:

## Demographic variables

▶ Ethnicity (9 categories, and numbers permitting, subdivision into 17 ONS groups).
▶ Sex.
▶ Age (continuous variable).
▶ Geographical region.

## Concurrent medication

▶ Glucose-lowering medications (metformin, sulfonylureas, thiazolidinedione, SGLT-2 inhibitors, DPP-4 inhibitors, GLP-1R agonists) and insulin therapy.
▶ Hormone therapy (in women).

## Clinical/blood test assessments

▶ Body mass index.
▶ Blood pressure (systolic and diastolic).
▶ Haemoglobin A1c.
▶ Total cholesterol and low-density lipoprotein.
▶ Liver function tests, including alanine aminotransferase.
▶ Estimated glomerular filtration rate.

## Comorbidities

▶ Sickle cell status.[10]
  – Sickle trait.
  – Sickle cell disease.
▶ Diabetes (type 1 and type 2 separately).[13 16]
  – Diabetes duration.
▶ Chronic kidney disease (CKD).
  – CKD stage 3.
  – CKD stage 4.
  – CKD stage 5 (including those requiring dialysis or ever receiving a transplant).
▶ Cancer.
  – Active or past haematological or solid cancer.
▶ Transplant (solid organ or bone marrow).
▶ Vaccination status.

## Key confounding variables
### Demographic variables

▶ Quintile of Townsend Deprivation Score: this is an area-level quintile score based on the patient's postcode. Originally developed by Townsend, it includes unemployment (as a percentage of those aged 16 and over who are economically active); non-car ownership (as a percentage of all households); non-home ownership (as a percentage of all households); and household overcrowding. These variables are measured for a given area of approximately 120 households, via the 2011 census, and combined to give a 'Townsend score' for that area. A higher Townsend score implies a greater level of deprivation.
▶ Residence type.
  – Lives in a care home (nursing home or residential care).
  – Homelessness.
  – Lives in own home.
▶ Household characteristics.
  – Age of household members.
  – Number of household inhabitants.

## Lifestyle factors

▶ Smoking status coded as non-smoker, ex-smoker, light (<10 cigarettes/day), moderate (10–19) and heavy (20+).

## Comorbidities

▶ Cardiovascular disease.
▶ Hypertension.
▶ Chronic respiratory disease.
▶ Liver cirrhosis.
▶ Depression.
▶ Dementia.
▶ Other chronic neurological diseases.
▶ Severe mental health illness.
▶ Learning disability.

## Concurrent medication

▶ Lipid-lowering medications.
▶ Antihypertensive medications.

All predictor variables will be based on the latest coded information recorded in the GP record prior to entry to the cohort.

### Sample size calculation

As of 9 August 2020, there had been 41 686 COVID-19 deaths in England out of a total population of 56 million. Assuming a 0.074% mortality rate in the general population, to detect a conservative estimate of an HR of 1.2 for COVID-19 mortality in the ethnic minority population that represents 18.6% of the study population (based on recent QResearch studies) with 90% power and a significance level of 0.01, we would need a sample size of 3 995 350 individuals with 2957 COVID-19 deaths. As recent studies of COVID-19 using QResearch have included cohorts of over 8 million adults with over 5000 COVID-19 deaths, even with 40% missing data, our study is sufficiently powered.

### Statistical methods

Participant characteristics will be summarised by ethnicity, age, sex, comorbidities and outcome with appropriate summary statistics. We will describe SARS-CoV-2 infection, COVID-19 hospitalisation and COVID-19 mortality rates in the general population and stratify this by ethnicity. Plots of survival curves stratified by ethnic group will be generated using the Kaplan-Meier method and compared with the log-rank test. We will use Cox proportional hazards models to estimate unadjusted and adjusted HRs for the associations between ethnicity and the outcomes of interest. When proportional hazards assumptions are not met, we will consider using more flexible models which do not assume proportionality of hazards, for example, flexible parametric survival models. Subsequently, we will seek to quantify the contribution of risk factor groups to

any observed increased mortality or hospitalisation risks in people of non-white ethnicity. For all analyses, we will conduct complete case analyses, and then separately impute for missing values, for example, ethnic group, body mass index, deprivation quintile and smoking status, using five imputations incorporating all model outcomes and predictors with model coefficients and SEs pooled in accordance with Rubin's rules. Data will be analysed using STATA V.16. Our study will be conducted and findings reported in line with the Strengthening the Reporting of Observational Studies in Epidemiology[17] and RECORD guidelines for observational studies using routinely collected health data.[18]

## Patient and public involvement

Patient representatives have been involved in the development of the study aims and design. We sought feedback from study inception and design from patients and the public within the Centre for BME Health, Leicester. We additionally have two public representatives as part of the study team whose contributions have been acknowledged in the authorship list of this manuscript. They will continue to be involved in all stages of the study, with additional public and patient views sought for dissemination strategies.

## Ethics and dissemination

The project has been independently peer-reviewed and received ethics approval from the QResearch Scientific Board (reference OX102). Findings will be disseminated through peer-reviewed manuscripts, presentations at scientific meetings and conferences with national and international stakeholders.

## DISCUSSION

Our findings aim to provide rapid evidence on the patterns of COVID-19 and associated mortality across and within ethnic groups in England. This study will be one of the largest COVID-19 cohorts, with a representative population across both rural and urban areas. Our cohort size will enable us to detect national-level variations within ethnic minority subgroups across a large geographical area and thus, to some extent, overcome selection bias limitations of previous cohort studies. Our findings have the potential to inform targeted mitigation public health strategies and could alter clinical thresholds for at-risk patients presenting with the infection.

As we are using routinely recorded clinical data from primary care records rather than prospectively collecting the data, we anticipate that our study will be subject to limitations. First, the precision and completeness of routine NHS patient records will vary, although previous work from QResearch suggests that key variables have high levels of accuracy and completeness. Second, our sample is likely to underestimate the total number of people in the population with COVID-19 due to the lack of systematic national testing programmes, which are still being rolled out, and also due to false-negative test results. There is also some evidence that testing rates are lower among ethnic minority groups. Our confirmed SARS-CoV-2 infection cases will additionally over-represent people with more severe disease who died or were hospitalised. During the earlier part of the pandemic, testing in the UK was limited to the hospital setting and people with COVID-19 who were not admitted or recovered (who may be from ethnic minority groups) are less likely to be captured in the data. Further, the admission criteria for Intensive Care Units varied during the course of the pandemic, with particular characteristics such as older age resulting in a lower chance of admission at a time when high admission rates were anticipated, which may introduce a selection bias.

**Author affiliations**
[1]Primary Care Research Centre, University of Southampton, Southampton, UK
[2]MRC Epidemiology Unit, University of Cambridge, Cambridge, UK
[3]Nuffield Department of Primary Care Health Sciences, University of Oxford, Oxford, UK
[4]Diabetes Research Centre, University of Leicester, Leicester, UK
[5]Division of Primary Care, University of Nottingham, Nottingham, UK
[6]PPI Representative, Leicester, UK
[7]Department of Health Sciences, University of Leicester, Leicester, UK
[8]Institute of Public Health, The Primary Care Unit, Cambridge, UK
[9]Nuffield Department of Primary Care Sciences, University of Oxford, Oxford, UK

**Contributors** HD-M led the study conceptualisation, wrote the first draft and revised the protocol. PST, DS and AKC critically revised the manuscript. FZ and CC critically revised the manuscript and contributed to the statistical methods. PL and FD critically revised the manuscript as PPI contributors. KK, SJG and JH-C contributed to study conceptualisation, study design and revised the protocol. The corresponding author attests that all listed authors meet the authorship criteria and that no others meeting the criteria have been omitted. HD-M is the guarantor.

**Funding** This work is funded by an MRC grant (MR/V027778/1). SJG is supported by an MRC Epidemiology Unit programme (MC_UU_12015/4). The University of Cambridge has received salary support in respect of SJG from the NHS in the East of England through the Clinical Academic Reserve. HD-M is a National Institute for Health Research (NIHR)-funded Academic Clinical Lecturer. JH-C receives support from the NHS and the NIHR and various research councils. KK and FZ are supported by the NIHR Applied Research Collaboration East Midlands (ARC EM) and the NIHR Leicester Biomedical Research Centre (BRC). The views and opinions expressed by authors in this publication are those of the authors and do not necessarily reflect those of the UK NIHR or the Department of Health and Social Care.

**Competing interests** JH-C is founder and director of QResearch database, which is a not-for-profit organisation with EMIS (leading commercial supplier of IT for 55% of general practices in the UK). JH-C is co-owner of ClinRisk and was a paid director until June 2019. ClinRisk develops open and closed source software to ensure the reliable and updatable implementation of clinical risk equations within clinical computer systems to help improve patient care, outside the submitted work. KK is national lead for NIHR ARC ethnicity and diversity, Director for the University of Leicester Centre for BME Health, Trustee of the South Asian Health Foundation and member of the Independent SAGE. The authors declare that no support from any organisation and no financial relationships have influenced the submitted work.

**Patient and public involvement** Patients and/or the public were involved in the design, or conduct, or reporting, or dissemination plans of this research. Refer to the Methods section for further details.

**Patient consent for publication** Not required.

**Provenance and peer review** Not commissioned; externally peer reviewed.

## ORCID iDs
Hajira Dambha-Miller http://orcid.org/0000-0003-0175-443X
Ashley Kieran Clift http://orcid.org/0000-0002-0061-979X
Kamlesh Khunti http://orcid.org/0000-0003-2343-7099
Julia Hippisley-Cox http://orcid.org/0000-0002-2479-7283

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
