## [Reviewer comments · BMJ Open]

ARTICLE DETAILS

TITLE (PROVISIONAL)	Quantifying the association between ethnicity and COVID-19 mortality: A national cohort study protocol
AUTHORS	Dambha-Miller, Hajira; Tan, Pui San; Saatci, Defne; Clift, Ashley; Zaccardi, Francesco; Coupland, Carol; Locufier, Patrick; Davies, Firoza; Khunti, Kamlesh; Griffin, Simon; Hippisley-Cox, Julia

VERSION 1 – REVIEW

REVIEWER	Christopher T. Rentsch London School of Hygiene & Tropical Medicine, US Department of Veterans Affairs
REVIEW RETURNED	13-Oct-2020

GENERAL COMMENTS	It was my pleasure to review Dr. Dambha-Miller and colleagues' protocol on examining ethnic disparities in COVID-19 positivity and severe outcomes in the UK. The authors set out to quantify the associations between ethnicity and COVID-19 outcomes in 18% of the UK population, allowing for precision in estimates for less sizeable ethnic groups. While this work is timely and important, I have a few concerns regarding the study design that I hope are addressable. I highlight my concerns here by section: Introduction (1) The OpenSAFELY paper by Williamson et al (citation #7) did not set out to explore ethnic disparities as the sole exposure variable. The OpenSAFELY pre-print by Mathur et al uploaded to medRxiv on 23 Sep 2020 (https://www.medrxiv.org/content/10.1101/2020.09.22.20198754v1) may be a more appropriate reference as it set out to perform largely similar analyses to those proposed by Dr. Dambha-Miller; however, the OpenSAFELY team were able to capture 40% of the population in England. I wonder if there is anything unique in the QResearch data or analytic plans that could build upon what has already been done in OpenSAFELY and elsewhere. Of course, QResearch can expand findings to the whole of the UK versus England only (which is needed and important!), but, for example, could the analyses take into account important interactions set a priori (e.g., ethnicity*age, ethnicity*sex, ethnicity*IMD, ethnicity*region)? Or look at risk of hospitalisation or death among positives only? Or death among hospitalised only? If there were any way to obtain occupation data, that would be hugely important to advance our understanding of ethnic disparities in COVID-19. (2) The paper by Price-Haywood et al (citation #14) is likely "contradictory" because it is not a nationwide sample, and only includes those who have tested positive. Therefore, collider bias may be present in those results (https://www.medrxiv.org/content/10.1101/2020.05.04.20090506v3).
---

You may want to consider instead relating findings at a population-level in the UK to those found in the largest integrated healthcare system in the US, the Veterans Health Administration (closest thing to NHS that US has). This paper was published in PLOS Medicine on 22 Sep 2020

(<https://journals.plos.org/plosmedicine/article?id=10.1371/journal.pmed.1003379>). It demonstrates that the excess risk in testing positive among racial and ethnic minorities translates to excess burden of deaths nationwide; however, when only analysing disparities in COVID-19 mortality among positives, there was no difference between race/ethnicity groups, which may be akin to what Price-Haywood found.

(3) Full disclosure: I am a co-author on the OpenSAFELY papers and lead author on the US paper in PLOS Medicine. I have no COI with the authors on this protocol, and I look forward to this work being carried out in QResearch. My intention in this review is to shed some light on the loads of feedback my teams have received during the publication process and public comments.

Methods. (1) Not clear where testing data is coming from. If SGSS, should comment on the lack of representativeness. If codes in primary care records, should say so. (2) The names and descriptions of any of the linked datasets are not adequately described. Importantly, the proportion of QResearch patients that can be linked to testing, hospitalisation, or death records should be stated. If not 100%, is the inability to link associated with ethnicity? (3) It is not clear whether 10 million records equate to 10 million patients or not. (4) The representativeness of the 18% of UK population in QResearch to the general population should be described, particularly in relation to ethnicity. (5) I assume 24 Jan 2020 is baseline for everyone in the study, but this is not specifically defined. If so, consider adjusting baseline ahead to something like 1 Mar 2020. The proportional hazards assumptions are unlikely to hold across January to March and beyond. (6) Mortality outcome. If using U07.1 and U07.2, consider specifying this. Also, will death records with these codes anywhere on certificate be counted as COVID-19 death, or will they need to be listed as primary cause of death? Same question with hospitalisation outcome: will a COVID-19 code anywhere on hospitalisation record be counted or does it need to be primary/admitting diagnosis? (7) Will ITU admission inherently "include patients receiving mechanical ventilation" or will you be able to identify those receiving mechanical ventilation separately? If the latter, consider including a separate outcome if powered. (8) Consider using negative control outcome (e.g., non-COVID-19 related mortality). As in, are the disparities seen in COVID-19 mortality specific to COVID-19? (9) Suggest listing the 17 categories of ethnicity in the protocol. (10) I found the definitions and specifications of variables lacking. At what time point/interval are each of the covariates being ascertained? Highly suggest building a study diagram to clearly show the depiction of study design (https://www.repeatinitiative.org/uploads/1/0/8/4/108495483/anninternermed.viz_study_design.2019.pdf). Are they based on diagnostic codes, laboratory measures, etc? If diagnostic codes, which code system and which codelists? How is geographic region defined? How will the levels of residence type be identified/defined? (11) I'm confused on the analytic approach; variables at different times throughout the protocol are referred to as "explanatory variables", "confounding variables", "predictor variables", "risk factors", and "mediators." Are these distinctions purposeful? Regardless, consider clarifying the analytic approach. (12) How will proportional hazards

	be checked and handled if found to be violated? (13) The reference for RECORD (citation #21) is incorrectly citing RECORD-PE. (14) Open science. Consider including a data sharing agreement. Highly recommend open coding practices if data cannot be shared.
--	--

REVIEWER	Gunnar Andersson Stockholm University Demography Unit Sweden
REVIEW RETURNED	01-Dec-2020

GENERAL COMMENTS	This protocol concerns a large-scale register-based study on the association between ethnicity and COVID-19 morbidity and mortality in the UK. The study is to be based on very rich large-scale data and the topic for research is indeed pressing; COVID-19 mortality differentials by ethnicity make some of the most intriguing observations during the 2020 pandemic. The authors rightly claim that population-based studies with individual-level data are badly needed. The proposed study will likely be the first of its kind in Britain but there are some studies based on population register data from Sweden to rely on for suggestive evidence. Still, the richness of data from Britain's national health care system has the capacity to produce entirely new insight into ethnicity-specific mortality differentials during the course of the pandemic. The methods and setup of the proposed analyses are entirely accurate for the task that is formulated. A set of plots of KM survivor curves may not be entirely lucid in their own right but produce univariate statistics as the basis for subsequent multivariate analyses. The variables with socio-demographic confounders are well chosen, but could perhaps be expanded with controls for factors such as educational attainment and marital status. The authors discuss various relevant cases of possible biases that may appear in the proposed study setup. These are all important aspects but I would still ask for some further information on the primary selectivity of people into the databases that will produce the exposures for COVID-19 morbidity and mortality. The reader would benefit from some information on the extent to which this study population is produced by different modes of health-seeking behavior or represents a more general population of people in England and Wales. There may also be some further information on the nature of missing ethnicity records. A sentence that claims that 40% missing data would be unproblematic seems less well chosen. The strategies with multiple imputations as a mode of sensitivity analysis was not entirely clear to me and could perhaps be spelled out a bit better.
--

VERSION 1 – AUTHOR RESPONSE

- **The OpenSAFELY paper by Williamson et al (citation #7) did not set out to explore ethnic disparities as the sole exposure variable. The OpenSAFELY pre-print by Mathur et al uploaded to medRxiv on 23 Sep 2020 (<https://www.medrxiv.org/content/10.1101/2020.09.22.20198754v1>) may be a more appropriate reference as it set out to perform largely similar analyses to those**

proposed by Dr. Dambha-Miller; however, the OpenSAFELY team were able to capture 40% of the population in England. I wonder if there is anything unique in the QResearch data or analytic plans that could build upon what has already been done in OpenSAFELY and elsewhere. Of course, QResearch can expand findings to the whole of the UK versus England only (which is needed and important!), but, for example, could the analyses take into account important interactions set a priori (e.g., ethnicity*age, ethnicity*sex, ethnicity*IMD, ethnicity*region)? Or look at risk of hospitalisation or death among positives only? Or death among hospitalised only? If there were any way to obtain occupation data, that would be hugely important to advance our understanding of ethnic disparities in COVID-19.

Thank you, we have included the suggested additional reference. We have added further clarity on what our study adds to the OpenSafely results including validation in a dataset that extends beyond England, the examination of interactions by ethnicity, household size, IMD, vaccination uptake etc; the introductory paragraph has been re-written.

- **The paper by Price-Haywood et al (citation #14) is likely “contradictory” because it is not a nationwide sample, and only includes those who have tested positive. Therefore, collider bias may be present in those results (<https://www.medrxiv.org/content/10.1101/2020.05.04.20090506v3>). You may want to consider instead relating findings at a population-level in the UK to those found in the largest integrated healthcare system in the US, the Veterans Health Administration (closest thing to NHS that US has). This paper was published in PLOS Medicine on 22 Sep 2020 (<https://journals.plos.org/plosmedicine/article?id=10.1371/journal.pmed.1003379>). It demonstrates that the excess risk in testing positive among racial and ethnic minorities translates to excess burden of deaths nationwide; however, when only analysing disparities in COVID-19 mortality among positives, there was no difference between race/ethnicity groups, which may be akin to what Price-Haywood found.**

Thank you we have amended the reference and text accordingly within the introduction.

- **Full disclosure: I am a co-author on the OpenSAFELY papers and lead author on the US paper in PLOS Medicine. I have no COI with the authors on this protocol, and I look forward to this work being carried out in QResearch. My intention in this review is to shed some light on the loads of feedback my teams have received during the publication process and public comments.**

Thank you for these comments. Your comments and learning from the OpenSafely study have been very helpful here.

- **Not clear where testing data is coming from. If SGSS, should comment on the lack of representativeness. If codes in primary care records, should say so.**

We have added additional text clarifying that this is from coding within the primary care records.

- **The names and descriptions of any of the linked datasets are not adequately described. Importantly, the proportion of QResearch patients that can be linked**

to testing, hospitalisation, or death records should be stated. If not 100%, is the inability to link associated with ethnicity?

As above and we have additionally provided the Q-Research database weblink which updates the extent of data linkage regularly. This will allow the reader to review the most up to date information on representativeness which is changing constantly.

- **It is not clear whether 10 million records equate to 10 million patients or not.**

Further clarity has been added to the text to explain this and additional references to the Q-Research database have been added. We have also referred the reader to the Q-Research website which regularly updates the numbers of patients who have died, moved area etc; This will allow the reader to have an up to date idea of exactly what the data represents on top of the text added to the protocol.

- **The representativeness of the 18% of UK population in QResearch to the general population should be described, particularly in relation to ethnicity.**

Additional text has been added including a link to the Q-Research website and we have clearly stated the sample represents 18% of the UK population. We have additionally included text on the current data which suggest >80% of ethnicity data has been recorded in the Q-Research population, although this might increase as the database is updated frequently. Where there are changes, we will report this fully in the main paper.

- **I assume 24 Jan 2020 is baseline for everyone in the study, but this is not specifically defined. If so, consider adjusting baseline ahead to something like 1 Mar 2020. The proportional hazards assumptions are unlikely to hold across January to March and beyond.**

Thank you for this helpful suggestion. We have included the 24th January as it is the first date of known COVID-19 infection in the UK, we have not included an end date as we will use all available data up until the time that we run the main analysis. This has already been stated in the text. In addition, when proportional hazards assumptions are not met, we will consider using more flexible models which do not assume proportionality of hazards e.g. flexible parametric survival models.

- **Mortality outcome. If using U07.1 and U07.2, consider specifying this. Also, will death records with these codes anywhere on certificate be counted as COVID-19 death, or will they need to be listed as primary cause of death? Same question with hospitalisation outcome: will a COVID-19 code anywhere on hospitalisation record be counted or does it need to be primary/admitting diagnosis?**

We have added the following text:

COVID-19 related mortality was defined as the presence of U07.1 or U07.2 on death certificates or a death from any cause in a patient with a confirmed positive SARS-CoV-2 test in the immediately preceding 28 days. COVID-19 hospitalisation was defined as hospitalisation with confirmed or suspected COVID-19 (as per ICD-10 codes), or new

hospitalisation in a patient with a positive SARS-CoV-2 test in the immediately preceding 14 days. We also refer to our published consensus statement on how these variables have been defined.

- **Will ITU admission inherently “include patients receiving mechanical ventilation” or will you be able to identify those receiving mechanical ventilation separately? If the latter, consider including a separate outcome if powered.**

This would be interesting to explore as the reviewer suggests but we don't have data to allow us to know if patients were mechanically ventilated, only that they were in ITU. Therefore, we have not altered the text here.

- **Consider using negative control outcome (e.g., non-COVID-19 related mortality). As in, are the disparities seen in COVID-19 mortality specific to COVID-19?**

This is an interesting approach that has been discussed with our statistical team. Where applicable, we will consider performing competing risk analysis to account for deaths due to non-COVID-19.

- **Suggest listing the 17 categories of ethnicity in the protocol.**

We have added this into the text.

- **I found the definitions and specifications of variables lacking. At what time point/interval are each of the covariates being ascertained? Are they based on diagnostic codes, laboratory measures, etc? If diagnostic codes, which code system and which codelists? How is geographic region defined? How will the levels of residence type be identified/defined?**

We have added in additional references that will direct readers to the diagnostic codes and codelists that have been previously published by Q-Research in relation to variables being used. Where possible, we have included these within the text. In general, we take first recorded dates for comorbidities, and most recent records for smoking, BMI. Read codes are used for GP diagnosis records, while ICD codes are used for hospital diagnosis records.

- **I'm confused on the analytic approach; variables at different times throughout the protocol are referred to as “explanatory variables”, “confounding variables”, “predictor variables”, “risk factors”, and “mediators.” Are these distinctions purposeful? Regardless, consider clarifying the analytic approach.**

We have added further text to ensure greater clarity in the analytical approach and also amended the text to ensure more consistency in the terminology used.

- **The reference for RECORD (citation #21) is incorrectly citing RECORD-PE. Open science. Consider including a data sharing agreement. Highly recommend open coding practices if data cannot be shared.**

We have updated the reference accordingly. A data sharing agreement has been included.

- **The methods and setup of the proposed analyses are entirely accurate for the task that is formulated. A set of plots of KM survivor curves may not be entirely lucid in their own right but produce univariate statistics as the basis for subsequent multivariate analyses. The variables with socio-demographic confounders are well chosen, but could perhaps be expanded with controls for factors such as educational attainment and marital status.**

Thank you for this suggestion. We agree that educational attainment and marital status are important. The marital status variable is not sufficiently robust to allow inclusion but we will interrogate the database further to try and include these additional variables in future work.

- **The authors discuss various relevant cases of possible biases that may appear in the proposed study setup. These are all important aspects but I would still ask for some further information on the primary selectivity of people into the databases that will produce the exposures for COVID-19 morbidity and mortality.**

Thank you. We agree this is important and have provided further information as outlined above the Q-Research database including the population characteristics, geographical inclusion and representativeness. We have also provided a direct link to the Q-Research database which updates this information regularly.

- **The strategies with multiple imputations as a mode of sensitivity analysis was not entirely clear to me and could perhaps be spelled out a bit better.**

Additional text has been added :

We will impute for missing values of variables with missing values e.g. ethnic group, BMI, deprivation quintile and smoking status using five imputations incorporating all model outcomes and predictors with model coefficients and standard errors pooled in accordance with Rubin's rules. (Reference: Rubin, D.B. (1987). Multiple Imputation for Nonresponse in Surveys. New York: John Wiley and Sons.)

VERSION 2 – REVIEW

REVIEWER	Christopher Rentsch London School of Hygiene & Tropical Medicine
REVIEW RETURNED	31-Jan-2021

GENERAL COMMENTS	The citation for evidence from OpenSAFELY on ethnic disparities in COVID-19 has been appropriately updated. However, the quoted estimates remain from the previous citation. The estimates should be updated to be in line with current citation.
---

	Instead of providing information on the data sources within the protocol, the authors choose to link to their website, which is entirely reasonable given the fast-changing nature of near real-time data to study COVID-19. However, they link they provided (https://www.qresearch.org/data/summary-of-gp-population-characteristics/#h-H2_0) raises further questions: (1) The website only describes data from England whilst the protocol highlights that a key feature of their proposed analysis is that it extends existing evidence to the whole of the UK. (2) Linked data for hospitalisations and deaths are reported to only extend through May 2019 or Dec 2017, respectively. Confusion remains around the baseline/index date for all patients in this study. The revised protocol states: "The index date will be the first recorded instance of the outcomes of interest." What is the index date to be used who do not experience the outcome?
--	---

REVIEWER	Gunnar Andersson Stockholms Universitet
REVIEW RETURNED	15-Feb-2021

GENERAL COMMENTS	The revision brought additional clarity to the study protocol. I look forward to seeing the study getting into the field.
---

VERSION 2 – AUTHOR RESPONSE

- **The citation for evidence from OpenSAFELY on ethnic disparities in COVID-19 has been appropriately updated. However, the quoted estimates remain from the previous citation. The estimates should be updated to be in line with current citation.**

We have provided the citation but removed specific hazard ratio estimates to avoid confusion as there are a number of different values that could be quoted depending on the ethnic group.

- **Instead of providing information on the data sources within the protocol, the authors choose to link to their website, which is entirely reasonable given the fast-changing nature of near real-time data to study COVID-19. However, they link they provided (https://www.qresearch.org/data/summary-of-gp-population-characteristics/#h-H2_0) raises further questions: (1) The website only describes data from England whilst the protocol highlights that a key feature of their proposed analysis is that it extends existing evidence to the whole of the UK. (2) Linked data for hospitalisations and deaths are reported to only extend through May 2019 or Dec 2017, respectively.**

We have revised the paper to describe England and not all of the UK. The Q-Research website has been updated in response to the reviewer comments. It now reads: "These figures are for illustration purposes since the database is regularly updated".

- **Confusion remains around the baseline/index date for all patients in this study. The revised protocol states: "The index date will be the first recorded instance of**

the outcomes of interest.” What is the index date to be used who do not experience the outcome?

The text now reads as follows ‘Study period: We will include a period from the date of the first case of COVID-19 in the UK (24th January 2020) until death, deregistration or the 30th October 2020’